

# The influence of inactivated entomopathogenic bacterium *Bacillus thuringiensis* on the immune responses of the Colorado potato beetle

Olga V. Polenogova[1], Natalia A. Kryukova[1], Tatyana Klementeva[1], Anna S. Artemchenko[1,2], Alexander D. Lukin[2], Viktor P. Khodyrev[1], Irina Slepneva[3], Yana Vorontsova[1] and Viktor V. Glupov[1]

[1] Institute of Systematics and Ecology of Animals, Siberian Branch of Russian Academy of Sciences, Novosibirsk, Russia
[2] Novosibirsk State University, Novosibirsk, Russia
[3] Voevodsky Institute of Chemical Kinetics and Combustion, Siberian Branch of Russian Academy of Sciences, Novosibirsk, Russia

Corresponding author
Olga V. Polenogova,
ovp0408@yandex.ru

## ABSTRACT

**Background:** Invasion of microorganisms into the gut of insects triggers a cascade of immune reactions accompanied by increased synthesis of effectors (such as antimicrobial peptides, cytokines, and amino acids), leading to changes in the physiological state of the host. We hypothesized that even an inactivated bacterium can induce an immune response in an insect. The aim of this study was to compare the roles of reactive oxygen species (ROS) formation and of the response of detoxification and antioxidant systems in a Colorado potato beetle (CPB) larval model in the first hours after invasion by either an inactivated or live bacterium.

**Methods:** The influence of *per os* inoculation with inactivated entomopathogenic bacterium *Bacillus thuringiensis* var. *tenebrionis* (Bt) on the survival and physiological and biochemical parameters of CPB larvae was assessed as changes in the total hemocyte count (THC), activity of phenoloxidases (POs), glutathione-S-transferases (GSTs), nonspecific esterases (ESTs), catalase, peroxidases, superoxide dismutases (SODs) and formation of reactive oxygen species (ROS).

**Results:** A series of changes occurred within the hemolymph and the midgut of CPBs inoculated with inactivated Bt at 12 h after inoculation. These physiological and biochemical alterations serve to mediate generalized resistance to pathogens. The changes were associated with an increase in the THC and a 1.4–2.2-fold enhancement of detoxification enzymatic activities (such as GST and EST) as well as increased levels of antioxidants (especially peroxidases) in hemolymph in comparison to the control group. Suppressed EST activity and reduced ROS formation were simultaneously detectable in the larval midgut. Inoculation of beetle larvae with active Bt cells yielded similar results (elevated THC and suppressed PO activity). A fundamental difference in the immune activation processes between larvae that ingested the inactivated bacterium and larvae that had consumed the active bacterium was that the inactivated bacterium did not influence ROS formation in the hemolymph but did reduce their formation in the midgut. At 24 h postinfection with active Bt, ROS levels went up in both the hemolymph and the midgut. This was accompanied by a significant 5.7-fold enhancement of SOD activity

and a 5.3-fold suppression of peroxidase activity. The observed alterations may be due to within-gut toxicity caused by early-stage bacteriosis. The imbalance in the antioxidant system and the accumulation of products toxic to the "putative" pathogen can activate detoxification mechanisms, including those of an enzymatic nature (EST and GST). The activation of detoxification processes and of innate immune responses is probably due to the recognition of the "putative" pathogen by gut epithelial cells and is similar in many respects to the immune response at early stages of bacteriosis.

## INTRODUCTION

Potato is the fifth most widely cultivated crop globally, with an area of approximately 17.8 million hectares, as indicated by the latest statistics from the Food and Agriculture Organization (FAO) (http://www.fao.org/faostat/en/#data/QC/visualize). A substantial challenge for potato and other nightshade crops is the presence of pests and diseases that have the potential to significantly impact yields (*Radcliffe & Lagnaoui, 2007*). One of the most important pests affecting potato crops is the Colorado potato beetle (CPB; *Leptinotarsa decemlineata* Say) (Coleoptera: Crysomelidae). Feeding stages of beetles (larvae and adults) can completely destroy the vegetative mass of the plants, thereby possibly resulting in the loss of up to 75% of the potato harvest (*Oerke, 2006*; *Sablon et al., 2013*). The CPB is a highly adaptive species with a high reproductive rate and migratory behavior, which has enabled it to become widespread (*Radcliffe & Lagnaoui, 2007*; *Alyokhin et al., 2013*). To date, the CPB has demonstrated resistant to more than 54 insecticides and has the capacity to attack genetically modified crops (*Molnar & Rakosy-Tican, 2021*).

The application of natural agents for the management of insect pests is a pertinent topic in the context of organic agriculture. Bacteria, fungi, and their metabolites (*Alyokhin et al., 2008*), as well as plant extracts (*Skuhrovec et al., 2015*) have emerged as key components in the development of ecofriendly insecticides. The best-known commercial bioinsecticides are based on soil-derived spore-forming bacteria of the species *Bacillus thuringiensis* (Bt) (*Berliner, 1915*; *Kumar, 2015*; *Singh, Bhardwaj & Singh, 2019*). It is widespread and plays an important role in the control of population of various insect species, including the CPB (*Kumar, 2015*; *Singh, Bhardwaj & Singh, 2019*). Nonetheless, the efficacy of Bt-based insecticide treatments may decrease due to fluctuations of environmental conditions (*e.g.*, reduced temperature, UV radiation, reduced rainfall, and duration of exposure) (*Ignoffo, 1992*; *Ferro, Slocombe & Mercier, 1997*; *Nault, Costa & Kennedy, 2000*), and this problem may require repeated applications (*Thakur et al., 2020*) and hence potential emergence of resistance in the CPB.

The insecticidal effect of Bt is attributable to the presence of crystalline endotoxins (Cry and Cyt toxins). Upon recognition by specific receptors located on the surface of the gut

epithelium, the endotoxins are activated under alkaline conditions in the midgut with subsequent pore formation and lysis of epithelial cells (*Bravo, Gill & Soberon, 2007*; *Melo, Soccol & Soccol, 2016*). These features allow Bt to serve as a basis for environmentally friendly bioinsecticides to control CPB populations (*Göldel, Lemic & Bažok, 2020*). The efficacy of these agents is especially pronounced against preimaginal stages, particularly juvenile stages (*Pucci, 1992*; *Wierenga, Norris & Whalon, 1996*; *Ghassemi-Kahrizeh & Aramideh, 2015*). Furthermore, these agents have been found to impede the development of imaginal stages of the CPB (*Nault, Costa & Kennedy, 2000*). The main toxic effects of Bt are due to endotoxins (Cry and Cyt) produced by the bacterium in the stationary phase, and in addition, there are soluble toxins (belonging to families Vip and Sip), which are secreted by vegetative cells and can also interact with specific receptors (*Malovichko, Nizhnikov & Antonets, 2019*; *Núñez-Ramírez et al., 2020*; *Gupta, Kumar & Kaur, 2021*). Aside from interacting with toxins, gut cells can also interact directly with the bacterial cell wall through certain pattern recognition receptors (*Wang et al., 2019*). This event may lead to increased synthesis of neurotransmitters, change the physiological state of the host, enhance enzyme synthesis in the gut, and consequently alter the structure of the insect gut microbiota (*Li et al., 2020*). In natural insect populations, acute infections are rare (*Buchon, Broderick & Lemaitre, 2013*; *Ryu et al., 2008*; *Guo et al., 2014*). Nonetheless, exposure to persistent opportunistic microorganisms can lead to the development of adaptive or acquired immunity: a phenomenon known as immune priming (*Sheehan, Farrell & Kavanagh, 2020*). The mechanism of priming in insects treated with an inactivated entomopathogenic bacterium is currently poorly understood (*Gomes et al., 2022*).

Insect resistance to pathogens is provided by a set of innate immune responses, various enzyme cascades, and the gut microbial community. Activation of an innate immune response and its regulation are complex, involving Imd and Toll signaling pathways and interactions within the hemocyte population as well as pathways of signaling and recognition of microbial components; these phenomena work together to regulate the immune response (*Commins, Borish & Steinke, 2010*; *Buchon, Silverman & Cherry, 2014*; *Sheehan, Farrell & Kavanagh, 2020*). Triggers of a cellular immune response are components of microbial cell walls and membranes as well as pathogen receptors of different types. These include (i) polysaccharides such as peptidoglycan, lipopolysaccharides, and glucans (*Kang et al., 1998*; *Swaminathan et al., 2006*); (ii) lipid-related compounds such as lipoteichoic acids (LTA) and lipoarabinomannan (*Ariki et al., 2004*; *Rao & Yu, 2010*); (iii) proteins and polypeptides such as flagellin and the capsid protein (*Hayashi et al., 2001*); and (iv) nucleic acids (*Oliveira, 2014*).

Recognition of microbial components by an insect launches cellular and humoral immune responses. Granulocytes and plasmatocytes play a crucial part in the identification and recognition of various pathogens through a set of pathogen-associated receptors present on their cell surface (*Jiang, Vilcinskas & Kanost, 2010*). These cells also exhibit an adhesive activity making them similar to mammalian neutrophils (*Renwick et al., 2007*; *Cho & Cho, 2019*). Induction of a cellular immune response (nodulation, encapsulation, phagocytosis, and melanization) occurs after activation of the Jak–Stat pathway and is

accompanied by a release of various biomolecules (that activate humoral immunity), such as antimicrobial and cytokine-like peptides, into the insect hemocoel and by triggering of the prophenoloxidase (proPO) cascade (*Mandrioli et al., 2003*; *Yang et al., 2015*). In *Galleria mellonella*, the proPO molecules are mainly presented in the oenocytoids (*Schmit, Rowley & Ratcliffe, 1977*), but it has also been reported that granulocytes (in *Bombyx mori*) can also express proPO (*Liu et al., 2013*).

Antimicrobial peptide are signaling molecules that are important for the maintenance of the host microbiota balance and its tolerance to oxidative stress (*Tjabringa et al., 2003*; *Wimley, 2010*; *Mergaert, 2018*; *Bai et al., 2021*). Coordinated functioning of IMD and Jak–STAT pathways, together with the Duox–ROS system, underlies regulation of the gut microbial community and development of resistance to pathogens (*Bai et al., 2020*). According to *Xiao et al. (2017)*, the Mesh–DUOX system modulates the level of reactive oxygen species (ROS), and this effect is directly linked to the degree of tissue toxicity caused by bacterial toxins (*Sajjadian & Kim, 2020*). Antioxidant enzymes mediate the detoxification of ROS. These enzymes include superoxide dismutase (SOD), catalase (Cat), peroxidase (Per), glutathione-S-transferase (GST), and nonspecific esterases (ESTs) (*Wang, Oberley & Murhammer, 2001*; *Dubovskiy et al., 2008*; *Deska, 2020*). DUOX generates the superoxide anion outside of the midgut wall cells. Metalloenzymes SODs catalyze dismutation of superoxide into hydrogen peroxide, which is then converted into water by Cat and Per (*Fridovich, 1995*; *Lambeth, 2004*). During melanogenesis in insects, semiquinones are formed from catecholamines. These are radical forms that can react with oxygen to form peroxide (*Komarov et al., 2005*).

The superoxide anion and hydrogen peroxide are potent oxidants that exhibit antimicrobial activity (*Ha et al., 2005*). Nevertheless, they can also cause tissue damage. Tissue damage by superoxide anion is observed when this radical is in excess, and this damage is accompanied by changes in enzymatic activity and can be caused by bacteria or their individual components, which can persist even after bacterial death and induce chronic tissue inflammation (reviewed by *Ramachandran et al., 2021*). For example, LTA, a component of the cell wall of gram-positive bacteria, can spontaneously bind to the cell surface (*Hummell & Winkelstein, 1986*) and lead to a release of prostaglandin E2 (PGE2) and lysosomal enzymes and to the production of the superoxide anion, affecting the integrity of animal tissues (*Card, Jasuja & Gustafson, 1994*). Immune stimulation by live and dead bacteria and their individual components (LTA and peptidoglycan) has been shown in both vertebrates and invertebrates, with different response rates (*Moesby et al., 2005*; *Lazado et al., 2010*; *Miyashita et al., 2015*; *Wen et al., 2019*). Furthermore, it bears noting that the majority of studies about the impact of ROS on the development of immune responses have been conducted in *Drosophila melanogaster* (*Carton, Poirié & Nappi, 2008*; *Buchon, Silverman & Cherry, 2014*; *Myers et al., 2018*). By contrast, studies on the CPB in this field have been relatively limited. Those that have been conducted have been focused on immune responses in the larval gut in response to fungal infections and toxicoses (*Kryukov et al., 2021*).

We hypothesized that even an inactivated bacterium can induce an immune response in an insect. The aim of this work was to compare the roles of ROS formation and of the

response of detoxification and antioxidant systems in a CPB larval model in the first hours after invasion by either an inactivated or live bacterium.

# MATERIALS AND METHODS

## Bacteria

The entomopathogenic bacterium *Bacillus thuringiensis* var. *tenebrionis* (Bt) (*morrisoni*; H8ab) was cultured on nutrient agar (Himedia, India) at 28 °C for 5 days. A microbiological smear stained with a 5% aqueous eosin solution showed a spore:crystal ratio of 1:1. To prepare a Bt spore-and-crystal suspension, 150 mM sterile saline (SS) was used, and cells in the suspension were washed twice (at 6,000 $g$ for 10 min) in an aseptic environment. The titer of the bacterial suspension was determined using a Neubauer hemocytometer. The entomopathogenic bacterial titer of Bt was $5 \times 10^7$ spores and crystals per milliliter. The suspension was divided into two parts. One part was autoclaved at 121 °C for 15 min, while the other one was left untouched. After microbiological seeding of the suspension on nutrient agar, the extent of Bt inactivation was assessed 3 days later.

## Insects and bioassay

The CPB *L. decemlineata* was collected from insecticide-free potato fields (Novosibirsk Oblast, Russia; 53°44′3.534″N, 77°39′0.0576″E). Beetle larvae were maintained at a constant temperature of 25 °C (±1 °C) on freshly cut potato shoots (*Solanum tuberosum*) under laboratory conditions under a natural photoperiod (approximately 16:8 light: dark). For the experiments, fourth-instar larvae (4–6 h postmolt) were placed in ventilated plastic containers, with 10 individuals per container, each containing a fresh potato shoot cushioned with moist cotton.

One milliliter of a suspension of the active (hereafter: Bt) suspension or inactivated (hereafter: ABt) bacterium was applied by fine spraying to the surface of a freshly cut potato sprout (3 g). After drying for 20 min at room temperature, the sprouts were placed into containers with larvae. The control group's feed was treated with SS instead of the bacterial suspension. Every day, larval's weight (mean ± SD) was measured, and their feed was replaced with untreated feed. The observation period was 5 days. At least 20 biological replicates (one replicate = 10 pooled larvae) for each treatment were set up for this bioassay.

## Samples prepared

At 12, 24, and 48 h post-treatment (pt), CPB larvae were cryo-anesthetized (at 4 °C). The larvae were punctured in the cuticle to collect hemolymph. The fat body and midgut tissues were dissected on ice, cleared, and washed twice in SS.

For total hemocyte count (THC) analysis, 10 µl of hemolymph was placed into 80 µl of ice-cold (4 °C) anticoagulation buffer (AC) (62 mM NaCl, 100 mM glucose, 10 mM EDTA, 30 mM sodium citrate, 3.26 mM citric acid, pH 4.6), containing 0.1 mM N-phenylthiourea (PTU).

For GST, EST, Cat, Per, and SOD activity assays, 10 µl of hemolymph was added to 10 µl of ice-cold (4 °C) 0.1 M phosphate buffered saline (PBS) containing 0.1 mM PTU or

without PTU for determination of phenoloxidase (PO) activity and ROS quantitation. Samples were centrifuged for 10 min at 500 $g$ at 4 °C; the supernatants were subjected to measurements.

Fat body and midgut tissues were collected into separate tubes with 100 µl of 0.1 M PBS containing 0.1 mM PTU for measuring enzymatic activity and without PTU for ROS quantification. Tissues were resuspended by Bandelin ultrasonic homogenization (3 s, 1 cycle). The supernatant (at 10,000 $g$ for 10 min at 4°C) was used for analyses.

## THC and PO

For determining the THC, the hemolymph samples were centrifuged (at 500 $g$ for 5 min), and hemocytes were washed three times with 80 µl ice-cold AC. HEPES buffer (140 mM NaCl, 5 mM KCl, 6 mM glucose, 10 mM HEPES, pH 7.2) was then added to the precipitate. Hemocytes were counted on a Neubauer hemocytometer, and the data were expressed as the total number of hemocytes per ml of hemolymph. Each treatment group included 10 biological replicates (one replicate = one individual).

PO activity was measured by the method proposed by *Ashida & Söderhäll (1984)* with modifications. Cell-free hemolymph (10 µl) was added to 200 µl of 10 mM L-DOPA. Dopachrome formation was measured at 490 nm for 45 min at 27 °C. A minimum of 17 biological replicates (one replicate = one larva) were used in each treatment group.

## Detoxication enzymes

GSTs and ESTs were quantified spectrophotometrically in 5 µl of cell-free hemolymph or in 15 µl of homogenized fat body and midgut tissues. Each treatment group included a minimum nine biological replicates (one replicate = a pool of two individuals). GST activity was measured at 340 nm after 15-min incubation with a substrate (2-nitro-5-thiobenzoic acid) at 28 °C according to *Habig, Pabst & Jakoby (1974)*. EST activity was measured at 414 nm after 15-min incubation with $p$-nitrophenylacetate at 28 °C according to the methods proposed by *Prabhakaran & Kamble (1995)*.

## Determination of the rate of ROS formation

Rates of ROS formation were determined in hemolymph and homogenates of midguts of CPB larvae at 12 and 24 h pt. The CP-H (1-hydroxy-3-carboxy-pyrrolidine) spin trap (*Dikalov, Skatchkov & Bassenge, 1997*; *Slepneva, Glupov & Khramtsov, 1999*) was used to measure the rates of formation of reactive intermediates in insect samples. CP-H is oxidized nonspecifically by highly oxidizing metabolites, resulting in the formation of a stable nitroxyl radical: CP. The time-dependent accumulation of the CP radical in the samples was studied by monitoring the amplitude of the low- field component of an electron paramagnetic resonance (EPR) spectrum.

CP-H was dissolved in oxygen-free (argon-bubbled) PBS-D (50 mM K, sodium phosphate buffer containing 50 µM diethylenetriaminepentaacetic acid (DTPA) and SS). DTPA was employed to diminish CP-H self-oxidation catalyzed by traces of transition metal ions. The mixtures of tested samples with 1 mM CP-H and 1 mM L-DOPA (for

hemolymph and without L-DOPA for midgut tissues) were placed into glass capillaries for EPR analyses.

EPR spectroscopy was performed at room temperature on an ER 200-D SRC X-band ESR spectrometer (Bruker). A minimum of 15 biological replicates (one replicate = two pooled larvae) were analyzed in each treatment group.

## Activity of antioxidant enzymes

Catalase (Cat), peroxidase (Per), and superoxide dismutase (SOD) activities were measured in cell-free hemolymph and midgut homogenates of CPB larvae at 24 h pt. Catalase activity was determined by means of the rate of hydrogen peroxide decomposition by a 5 µl sample in 195 µl of reaction mixture at 240 nm after 60 s incubation according to the procedure proposed by *Wong et al. (1991)*. Peroxidase activity was measured by the method proposed by *Nicell & Wright (1997)* with modifications. A 10 µl sample was added to a 100 µl reaction mixture consisting of 0.1 M PBS, 1.7 mM hydrogen peroxide, and 2.5 mM 4-aminoantipyrine. Optical density was measured at 510 nm after 4-min incubation at 25 °C. SOD activity was determined judging by the rate of reduction of nitroblue tetrazolium by the superoxide anion that is produced during oxidation of xanthine by xanthine oxidase (*McCord & Fridovich, 1969*). A total of 0.2 µl of each sample was added to a 200 µl reaction mixture composed of 0.75 mM xanthine and 0.3 mM nitro blue tetrazolium in 0.1 M PBS (pH 7.2) and 8 µl of 0.5 U xanthine oxidase. Xanthine oxidase was dissolved in 0.1 M PBS supplemented with 0.005% of bovine serum albumin. SOD concentration was determined at 560 nm after 30 min incubation at 27 °C. The unit of SOD activity was calculated as the difference between the amount of reduced nitroblue tetrazolium without participation of SOD and the amount of nitroblue tetrazolium reduced during inhibition by SOD for 1 min in 1 ml of the reaction solution, as calculated per mg of protein in the sample, U = (µM nitroblue tetrazolium/mg protein per min). Each treatment group included 12 biological replicates (one replicate = two pooled individuals).

Protein concentrations in the hemolymph and in fat body and midguts homogenates were determined by the Bradford method (*Bradford, 1976*). Bovine serum albumin was used as a standard for plotting the calibration curve.

The PO, EST, GST, Cat and Per activities in the reaction mixture during the assays were evaluated in transmission density units ($\Delta\alpha$) per min (per s for catalase) per mg of protein. The enzymatic activity was defined as an increase of 0.001 in absorbance per min per mg of protein.

## Statistical analysis

The Kaplan–Meier test was performed to analyze the survival and to build the curve of cumulative proportion of insects surviving in the experiment. Weight gain was expressed in g (mean ± SD). Statistical significance of non-normally distributed data (Shapiro–Wilk test, $P < 0.05$) was assessed by Kruskal–Wallis one-way analysis of variance followed by Dunn's *post hoc* test for multiple comparisons. Data were analyzed in GraphPad Prism v.4.0 (GraphPad Software Inc., Boston, MA, USA), Statistica 8 (StatSoft Inc., Tulsa, OK, USA), and PAST 3 software (*Hammer, Harper & Ryan, 2001*).

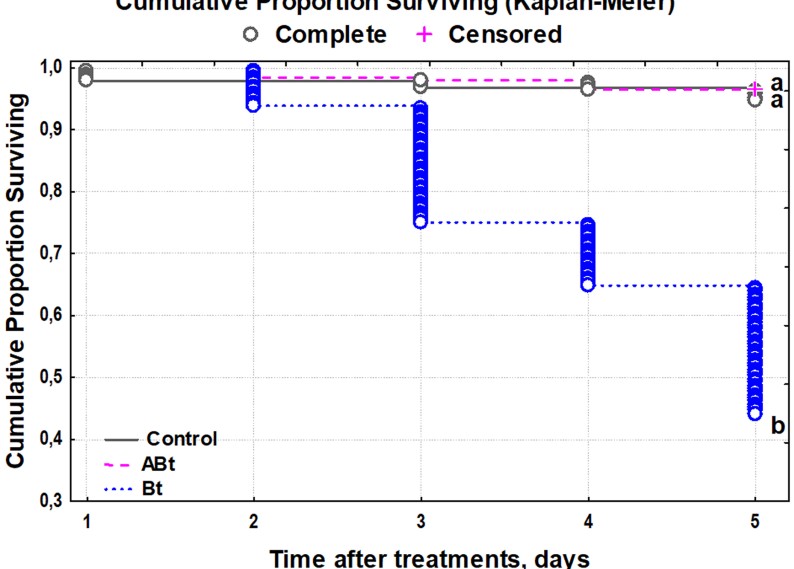

**Figure 1 The survival of CPB larvae after *per os* treatment with suspensions of active (Bt) and inactivated (ABt) entomopathogenic bacterium *B. thuringiensis* ($5 \times 10^7$ spores and crystals/mL suspension).** At least 60 individuals were used for each treatment group. The data from each time point was subjected to separate analysis. Significant differences between treatments (a and b) were identified by applying the Kruskal–Wallis test with subsequent Dunn's test ($P < 0.05$).

## RESULTS

### Bioassay and larval body weight of the CPB

The survival rate of the CPB was not affected by feeding of the ABt suspension, and the larvae had a survival rate of 99% (Fig. 1). By contrast, Bt infection reduced the survival starting on day 2 and onwards, with only a 37% survival rate by day 5 of observation (Kaplan–Meier test, $P < 0.001$, compared to other treatments group).

The body weight of larvae did not change after exposure to ABt throughout the observation period as compared to the control group ($P > 0.15$; Fig. 2). On the contrary, Bt infection significantly reduced the body weight of CPB larvae throughout the observation period (Dunn's test, $P < 0.05$, compared to the other group).

### THC and PO activity

A total of 12 h after *per os* administration of either Bt or ABt, there was a 1.3-fold increase in the number of hemocytes in the hemolymph (THC) of CPB larvae ($P < 0.005$, compared to the control; Fig. 3A). Nonetheless, 24 h after ABt exposure, THC levels did not differ from those in the control group ($P = 0.49$). Meanwhile, 24 h after Bt exposure, THC levels in the hemolymph of CPB larvae were significantly different from those in the control group ($P = 0.003$).

*Per os* inoculation of larvae with ABt reduced PO activity in CPB hemolymph at all time points, but significant differences were recorded 48 h after exposure ($P = 0.003$, compared to the control; Fig. 3B). In the treatment group where the suspension of active Bt cells was

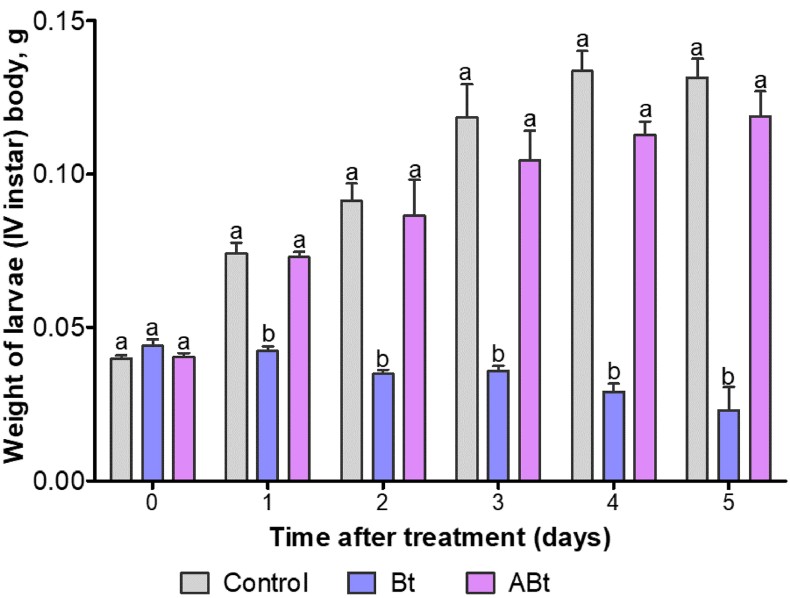

**Figure 2** **The dynamics of body weight of CPB larvae after *per os* treatment with suspensions of active (Bt) and inactivated (ABt) entomopathogenic bacterium *B. thuringiensis*.** Sixty individuals were used in each variant. The data from each time point was subjected to separate analysis. Significant differences between treatments (a and b) were identified by applying the Kruskal–Wallis test with subsequent Dunn's test ($P < 0.05$).

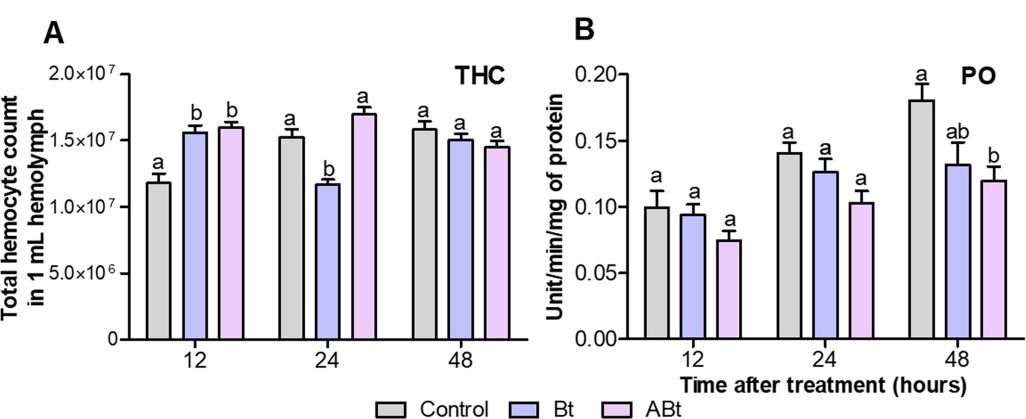

**Figure 3** **The total hemocyte count (THC) (A) and phenoloxidase (PO) activity (B) of CPB larvae 12, 24 and 48 h after *per os* treatment with suspensions of active (Bt) and inactivated (ABt) entomopathogenic bacterium *B. thuringiensis*.** Each treatment group included eight biological replicates for THC and twenty, for PO (one replicate = one individual). The data from each time point was subjected to separate analysis. Significant differences between treatments (a and b) were identified by applying the Kruskal–Wallis test with subsequent Dunn's test ($P < 0.05$).

administered *per os* to CPB larvae, there were no differences from the control group of insects in the PO activity indices ($P > 0.06$).

## Detoxifying enzymes

Detoxifying-enzyme assays revealed that GST and EST activity in the hemolymph of CPB larvae rose during the first 12 h after *per os* administration of the ABt or Bt suspensions

(Figs. 4A and 4B). After ABt feeding, GST activity increased 2.3-fold and EST activity increased 1.2-fold (valid for GST: $P = 0.002$, compared to the control). Bt infection was found to be associated with a 1.2-fold increase in GST activity and a 1.4-fold increase in EST activity (significant for EST: $P = 0.03$, compared to the control). Additionally, analysis of GST and EST activities in CPB hemolymph after feeding with ABt suspensions showed no significant differences in the parameters from the CPB control group, except for a 1.2-fold increase in EST activity at 48 h pt ($P = 0.03$). The results were somewhat consistent with the assays of GST and EST activities in CPB hemolymph following Bt infection. After 24 h of exposure to active Bt, GST and EST activities in the hemolymph of CPB larvae diminished (significant for GST at 24 h: $P = 0.04$, compared to control).

There were no significant differences in GST and EST activities within the fat body of larvae between the ABt suspension inoculation group and the control group at all time points (Figs. 4C and 4D). By contrast, Bt infection at 12 h pt resulted in 1.4-fold suppression of GST activity and a 1.3-fold increase in EST activity in the fat body of larvae (for both: $P < 0.001$, compared to control). Additional measurements of enzymatic activity in the fat body of CPB revealed that the infection led to a 1.5-fold decline of GST activity at 24 and 48 h (significant for time points 24 and 48 h: $P < 0.02$) and a 2-fold enhancement of in EST activity as compared to control CPB larvae (significant for the 48 h time point: $P = 0.04$) during the same period.

After only 48 h of exposure to ABt, enzymatic activity in the midgut of CPB larvae diminished 1.3-fold for both GST and EST (EST: $P = 0.001$, compared to the control, Figs. 4E and 4F). Throughout the development of Bt infection, enzymatic activity proved to be significantly suppressed (1.3- to 1.4-fold) at all time points (GST: $P < 0.01$, EST: $P < 0.05$, compared to the control).

## ROS production

Examination of ROS production by EPR spectroscopy showed that ABt feeding to the insects did not significantly increase the ROS level in the hemolymph of CPB larvae during 24 h ($P < 0.05$; as compared to the control, Fig. 5A). On the contrary, reduced ROS levels were observed in the midgut at both 12 and 24 h after feeding with the inactivated bacterium (significant for ABt at 12 h: $P < 0.000$, compared to the control, Fig. 5B). After Bt infection, ROS levels in the hemolymph of CPB larvae rose significantly ($P = 0.008$, compared to the control; Fig. 5A). The levels continued to increase over the next 24 h, thereby exceeding the control ones 3.5-fold ($P < 0.000$). The ROS levels in the midgut of CPB larvae decreased slightly ($P = 0.07$) at 12 h after Bt infection. Nonetheless, at 24 h post-infection, the ROS level was four-fold higher than that in control insects ($P < 0.001$; Fig. 5B).

## Antioxidant systems

No significant changes in activity of antioxidant enzymes (Cat, Per, and SOD) were detectable in the hemolymph and midgut of CPB larvae at 24 h pt with ABt suspensions (see Fig. 6). In contrast, Bt infection led to a 1.19-fold increase in Per activity in the hemolymph of CPB larvae ($P = 0.02$, compared to the control, Fig. 6B). Furthermore, after

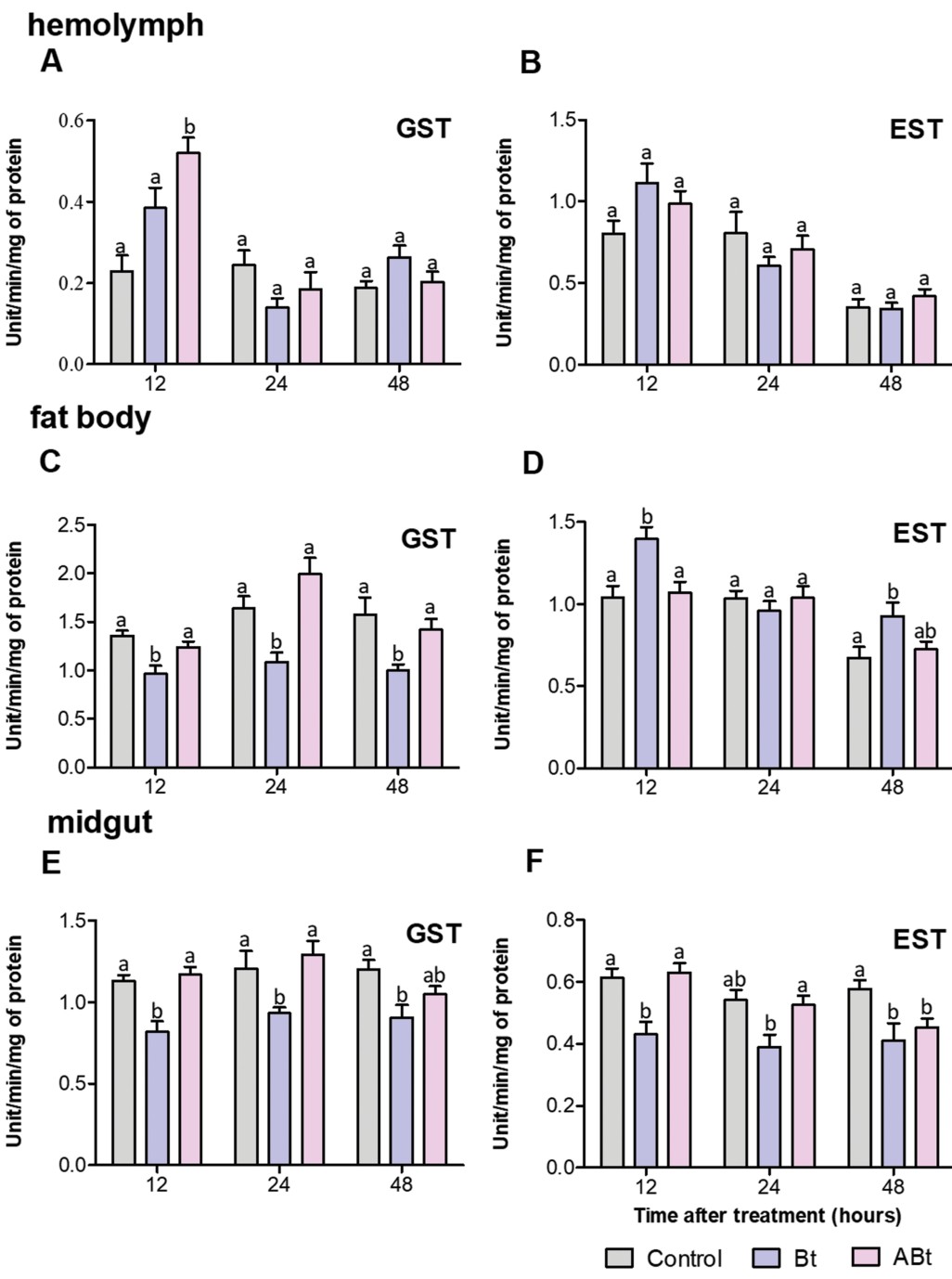

**hemolymph**

**fat body**

**midgut**

**Figure 4  Activity of glutathione-S-transferase (GST; A, C, E) and nonspecific esterase (EST; B, D, F) in the hemolymph, fat body and midgut of CPB larvae 12, 24, and 48 h after *per os* treatment with suspensions of active (Bt) and inactivated (ABt) bacterium.** Each treatment group included a minimum of 10 biological replicates (one replicate = two pooled individuals). The data from each time point was subjected to separate analysis. Significant differences between treatments (a and b) were identified by applying the Kruskal–Wallis test with subsequent Dunn's test ($P < 0.05$).

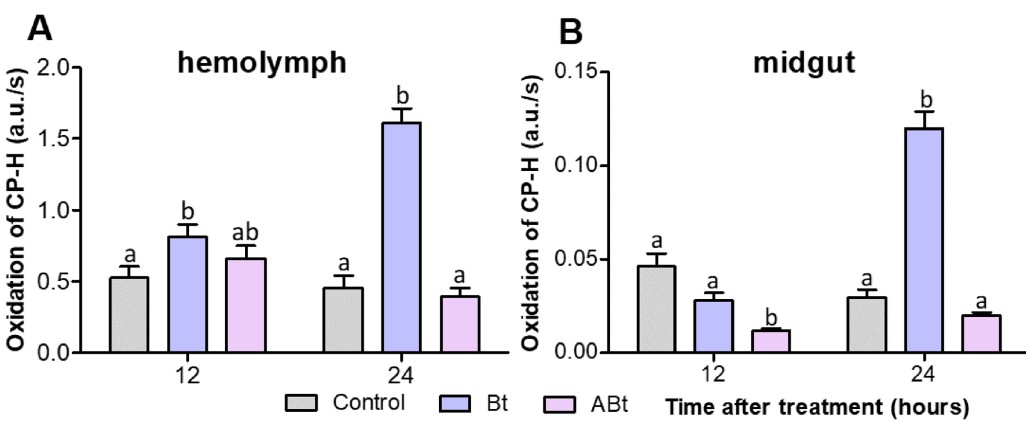

**Figure 5 ROS production in the hemolymph (A) and midgut (B) tissues of CPB larvae 12 and 24 h after *per os* treatment with suspensions of active (Bt) and inactivated (ABt) entomopathogenic bacterium *B. thuringiensis*.** Each treatment group included a minimum of 15 biological replicates (one replicate = two pooled individuals). The data from each time point was subjected to separate analysis. Significant differences between treatments (a and b) were identified by applying the Kruskal–Wallis test with subsequent Dunn's test (*P* < 0.05).

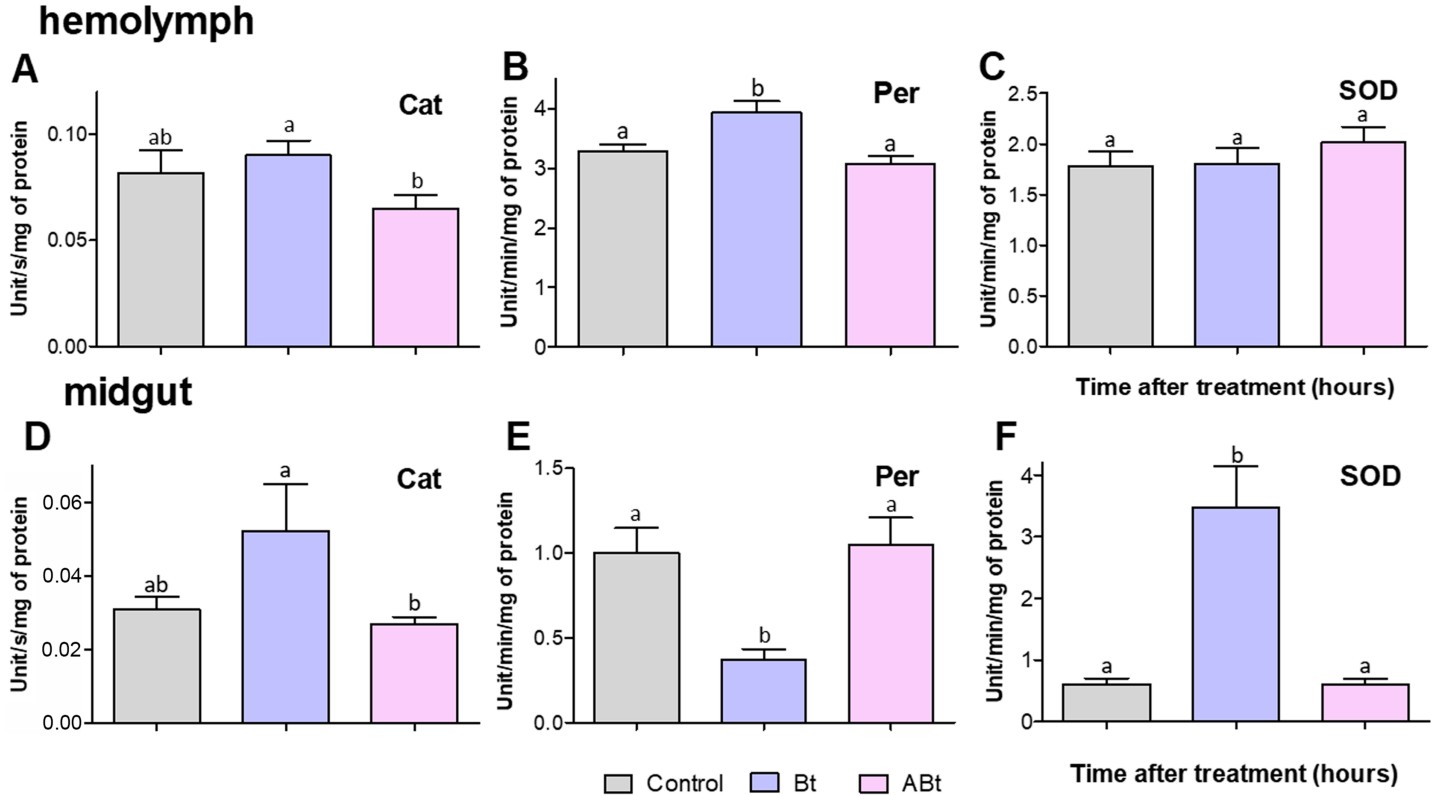

**Figure 6 Activity of peroxidase (Per; A and D), catalase (Cat; B and E) and superoxide dismutase (SOD; C and F) in the hemolymph (A–C) and the midgut (D–F) of CPB larvae 24 h after *per os* treatment with suspensions of active (Bt) and inactivated (ABt) bacterium.** Each treatment group included at least 11 biological replicates (one replicate = two pooled individuals). The data from each time point was subjected to separate analysis. Significant differences between treatments (a and b) were identified by applying the Kruskal–Wallis test with subsequent Dunn's test (*P* < 0.05).

administration of active Bt, there was a significant 5.7-fold increase in SOD activity and 5.3-fold suppression of Per activity in the midgut after 24 h (significant for SOD and Per: $P < 0.001$, compared to the control; Figs. 6E and 6F). Additionally, Cat activity showed marginally significant 1.3-fold enhancement ($P = 0.06$, Fig. 6D).

## DISCUSSION

Our experiments indicate that an immune response in CPB larvae got activated within the first 12 h after feeding with inactivated cells of the entomopathogenic bacterium Bt. The insect guts showed suppressed EST activity and ROS generation (up to time point "48 h"), while the hemolymph elevated activity of detoxifying enzymes (GST and EST). Inoculation of CPB larvae with active Bt bacterial cells had in a similar effect. This was accompanied by an increase in THC and suppression of PO activity in hemolymph. Nonetheless, a fundamental differences between the immune activation processes observed when inactivated or active Bt was administered was that the inactivated bacterium did not influence the production of ROS in hemolymph but rather reduced their formation in the midgut.

Overall, when the larvae were inoculated with the inactivated bacterium *via* oral administration, there was no significant reduction in their survival rate or any noticeable weight changes during the observation period (see Figs. 1 and 2). Meanwhile, as a result of Bt infection, the body weight of the CPB larvae decreased throughout the observation period. The weight of CPB larvae declined after the infection initiation owing to nutritional disturbances caused by toxicity of bacterial components and by the development of bacteriosis.

One potential mechanism of tissue damage in naturally occurring bacterial infections is the susceptibility of intrinsic defense proteins—that are constantly present in the hemolymph—to damage. In particular, it has been empirically demonstrated in mammals that both the bacterium *Bacillus subtilis* bacteria and individual bacterial cell components (LTA) are capable of inducing immune responses (*Card, Jasuja & Gustafson, 1994*; *Moesby et al., 2005*). LTAs have the capacity to form spontaneous bonds with the surface of cells (*Hummell & Winkelstein, 1986*), thus subsequently leading to the release of prostaglandin E2 (PGE2), lysosomal enzymes, and superoxide anions, thereby affecting the integrity of animal tissues (*Card, Jasuja & Gustafson, 1994*). Furthermore, certain bacterial pathogen-associated molecular patterns (*e.g.*, LTA) can persist even after cell death, thus leading to the onset of inflammatory processes in tissues (*Ramachandran et al., 2021*, in review). Findings in several studies indicate that not all pathogen-associated molecular patterns can be eliminated. For example, terminal sterilization by wet and dry steam (160 min at 134 °C) was found to be ineffective in ablating pyrogenic activity of LTA and peptidoglycan of *Staphylococcus aureus* and *B. subtilis* endospores (160 min at 134°C) (*Moesby, Timm & Hansen, 2008*). Furthermore, the autoclaving of *Bacillus* spores at 120 °C for 15 min does not result in any observable alterations of spore ultrastructure (*Kulikovskii, 1979*). One work suggests that endotoxin of Bt is thermolabile (*Lacey, Mulla & Dulmage, 1978*). In our experiment, however, light microscopy did not reveal significant destruction of crystal inclusions. Nonetheless, we do not rule out the possibility that

autoclaving caused partial fragmentation of protein structures of Bt crystals with loss of their insecticidal activity. This supposition requires more careful analysis by methods more sensitive than classical microscopy. Inactivated cells of *B. subtilis* (a species related to Bt) and *S. aureus* can induce cytokine production in vertebrates, albeit to a lesser extent than live bacterial cells can, as reported by *Moesby et al. (2005)*. Another example is the capacity of killed *B. subtilis* spores to function as a mucosal adjuvant for the H5N1 influenza vaccine, resulting in augmented both humoral and cellular immune responses in mice (*Song et al., 2012*). Furthermore, the administration of killed spores alone had a partial protective effect (60%) in animals (*Song et al., 2012*). Besides, research suggests that bacterial thermal inactivation does not significantly alter the effectiveness of certain probiotic strains. Of note, this process may actually enhance an innate immune response in fish (*Díaz-Rosales et al., 2006*; *Salinas et al., 2006*). It is possible that the Cry toxin of Bt also retains some of its structure after autoclaving, and it is highly probable that this toxin will retain the ability to bind to the host gut epithelium and to cause oxidative stress and tissue damage following terminal sterilization. The study by *Dubovskiy et al. (2021)* suggests that the *per os* administration of native Bt Cry3A toxins of Bt to CPB larvae induced oxidative stress and causes damage to midgut tissues, as evidenced by a 1.5-fold increase in ROS and malondialdehyde (lipid peroxidation) levels. This finding is consistent with our data about the feeding of active Bt cells to CPB larvae: enhancement of activities of detoxifying enzymes (GST and EST) along activities of peroxidases. Additionally, an augmentation of the production of ROS in hemolymph was observed. EST and GST participate in the detoxification of toxic substances in the body, the regulation of oxidative stress, and the sequestration of pathogens (*Lange et al., 2018*; *Ma et al., 2018*; *Zhou et al., 2019*). Furthermore, administration of sublethal doses of Cry3Aa crystals from active Bt to CPB larvae has been reported to result in weight loss and enhanced immunity (*García-Robles et al., 2020*). This phenomenon was also observed in our experiments after the inoculation of CPBs with active Bt cells. By contrast, the results obtained when ABt was fed to CPB larvae revealed no weight loss in individuals (Fig. 2) and an enhancement of detoxification processes in hemolymph during the initial hours. The initial period after pathogen recognition is of critical importance for host defense and is accompanied by the synthesis of AMP and is release into the hemolymph. The synthesis of AMP molecules is detectable after a period of 2–4 h and is suppressed by the end of the first day in order to maintain control values (*Uvell & Engström, 2007*). This was demonstrated for acaloleptin A and attacin B transcripts after poisoning of CPB larvae with a nonlethal dose of a mixture of Cry3Aa spores and crystals (*García-Robles et al., 2020*). It seems plausible that the recognition of an invading pathogen or its individual components in the gut is touch off processes of innate immunity in insects, even in the absence of gut damage. Nevertheless, further experimentation is required to substantiate this claim.

Our analysis of immune reactions revealed significant changes within the first 12 h after the feeding of CPB larvae on either inactivated or active bacterial cells, with the THC and PO activity being affected. Feeding on either ABt or Bt had similar effects, with an increase in the THC and a slight decrease in PO activity. By the end of the first day, however, the THC did not differ between the ABt group and the control group of larvae. Furthermore,

PO activity significantly diminished after 48 h. Meanwhile, the use of active Bt significantly reduced THC by the end of the first day of the experiment. The increase in the THC is likely to be caused by activation of the intestinal immunity during pathogen recognition and identification. This process is regulated by two signaling pathways, Imd and Toll, and is accompanied by activation of the Jak–Stat pathway. During these processes, modulators are released into the hemocoel and new hemocytes are either formed from pericardial bodies or released from depots (*Wu, Xu & Yi, 2016*; *Pastor-Pareja, Wu & Xu, 2008*; *Yang & Hultmark, 2016*). Our results suggest that the THC increases soon after initiation of infection with either active or inactivated Bt. This is probably due to the recognition of conserved microbial components or their combinations by pathogen-associated receptors on hemocytes or on the gut epithelium, and this event triggers a cellular immune response (*Bellocchio et al., 2004*; *Renwick et al., 2007*; *Dubovskiy, Krukova & Glupov, 2008*).

Probably, the method of bacterial inactivation employed by us preserves most of the structures of bacterial cell wall components, which allow the host to quickly recognize and identify potential pathogens. Several studies have shown that inactivated entomopathogens or their sublethal doses can increase hemocyte numbers during immune reaction priming (*Fallon, Troy & Kavanagh, 2011*; *Hernández López et al., 2014*; *Wu, Xu & Yi, 2016*). We found that an elevated GST level in the hemolymph directly correlated with higher hemocyte counts, after infection with either the active and inactivated bacterium. This correlation is likely to be due to direct involvement of GST in the synthesis of eicosanoids (*Hayes, Flanagan & Jowsey, 2005*; *Stanley, 2006*). These compounds actively stimulate the cellular immunity and, among other factors, take part in proliferation and release of hemocytes from the depot (*Merchant et al., 2008*; *Park & Kim, 2012*).

The immune response in insects is often measured by means of the activity of the phenoloxidase system in their tissues, including hemolymph. Our article indicated that oral administration of Bt, as either active cells or inactive suspensions, can inhibit activity of PO in the hemolymph. Microbiological composition of the gut may become imbalanced, thus reducing the number of symbiotic bacteria and negatively affecting the larval humoral immunity. *Lin et al. (2020)* detected a similar effect when *Plutella xylostella* larvae were infected with *B. thuringiensis* strain Bt8010. Additionally, Bt toxins may suppress PO activity during infection with a live spore–crystal mixture (*Grizanova et al., 2014*). The acute phase of Bt bacteriosis occurs on the second day after exposure. This phase is usually accompanied by pore formation in the intestine, hemocyte aggregation on the intestinal surface, and activation of phagocytosis and granuloma formation processes (*Zhang et al., 2014*; *Santhoshkumar et al., 2021*). These processes are likely to be responsible for the observed decrease in the THC levels at 24 h after Bt inoculation. We have previously obtained similar results in CPBs after *per os* infection by Bt and reintroduction of their symbiotic bacteria, *Citrobacter freundii*. Bt bacteriosis increased the number of free-circulating hemocytes and reduced the number of PO-positive cells in the hemolymph of CPBs; the changes took place within the first few hours after pathogen invasion (*Polenogova et al., 2022*). An increase in the THC is a notable indicator of insect immunity activation. A study published in 2022 by *Prabu et al. (2022)* indicates that the interaction of the Cry1F toxin with C-type lectins on the surface of hemocytes results in the

launch of the mitogen-activated protein kinase (MAP) pathway. These kinases directly initiate a number of cellular processes, including hemocyte aggregation, apoptosis, cytokinesis, and proliferation (*Pelech & Söderhäll, 1996*; *Goberdhan & Wilson, 1998*; *Johnson & Lapadat, 2002*). Therefore, the observed increase in the THC at 12 h after inoculation of Bt (either active or inactivated) may be indicative of enhanced cellular immunity resulting from the recognition of the toxin (Cry) or its fragments (in the case of inactivated Bt) by hemocytes.

We analyzed the activity of detoxifying and antioxidant systems, as well as ROS levels both in the hemolymph and the midgut of CPB larvae. Our results indicated that ABt led to a spike in GST activity in the hemolymph and a slight increase in the ROS level in the midgut within the first 12 h after exposure. Specific interaction between surface patterns of spores and crystals may be the reason; this is because their structure is identical to that of toxin antigens and they remain unchanged even after heat treatment (*Delafield, Somerville & Rittenberg, 1968*; *Gerhardt, Pankratz & Scherrer, 1976*; *Du & Nickerson, 1996*). The interaction of these patterns with receptors in gut cells may trigger immune responses similar to those caused by the active mixture of spores and crystals but to a lesser extent. On their surface, pathogens have structures that are recognized by an immune system and can be composed of different materials such as polysaccharides and lipids (such as LTA), *etc.*, *Ariki et al. (2004)*, *Swaminathan et al. (2006)*, *Parusela et al. (2017)*, and these structures may be resistant to high temperatures without being destroyed.

We hypothesized that pathogen recognition in the gut is sufficient to trigger the activate processes of the innate immune response, and that the intestinal epithelium and gut microbiota are only secondary participants in this process. Our data show that EST activity went up increased after ABt feeding (at 48 h), thereby supporting this hypothesis. In addition, feeding of inactivated Bt and active Bt cells had some similar effects. Bt infection caused an increase in the levels of ROS, antioxidant, and detoxification systems in hemolymph. EST and GST promote detoxification of toxic products in the body, regulate oxidative stress, and sequester pathogens (*Lange et al., 2018*; *Ma et al., 2018*; *Zhou et al., 2019*). It is proposed that this phenomenon is primarily associated with the recognition by the native microbiota or gut epithelium of inactivated Bt cells or of their individual components that may have persisted after cell death and may result in the synthesis of immune-response mediators. Such alterations in the midgut may represent a mechanism ensuring microbial homeostasis in the insect gut during feeding or during the development of bacterial infections (*Oliveira et al., 2011*; *Sajjadian & Kim, 2020*). Furthermore, these mediators are capable of penetrating the hemocoel and of activating both cellular and humoral immunity, which is accompanied by the release of free-radical oxygen species and AMPs into the hemolymph. In a study conducted by *Freitak et al. (2014)*, it was demonstrated that non-pathogenic bacteria, when administered *per os* to *G. mellonella* larvae, results in the activation of the PO system and lysozyme in hemolymph. This was accompanied by the induction of some AMPs.

Our findings may indicate that the activation of detoxification processes and innate immune responses is probably due to the recognition of a "putative" pathogen by gut

epithelial cells and is similar in many respects to the immune response seen at early stages of bacteriosis.

## CONCLUSIONS

The study suggests that when administered orally, either an inactivated or active bacterium induces alterations in the levels of THC, PO activity, detoxifying and antioxidant enzymes, and ROS production in CPB larvae. It seems plausible that in CPBs, the systemic immune response observed after the recognition of a "putative" pathogen both by gut epithelial cells and by the microbiota may be due to the native microbiota's secondary metabolites that are capable of crossing the gut barrier and penetrating the hemocoel without binding to the gut epithelium or causing tissue destruction. We hypothesized that the obtained results are related to the activation of these mechanisms. Nevertheless, additional research is necessary to gain a deeper comprehension of these mechanisms, in particularly the influence of the native microbiota on immune-response development and on free-radical oxygen species synthesis after the ingestion of inactivated Bt and/or its components by CPB larvae; this issue can be effectively addressed through transcriptomic and genomic experiments.

## ACKNOWLEDGEMENTS

The authors are grateful to Dr. V. A. Shilo (Karasuk Station of the Institute of Systematics and Ecology of Animals SB RAS) for help in organizing the experiments. The English language was corrected by Shevchuk Editing.

### Funding
This research was funded by the Russian Science Foundation (No. 22-76-10051). The funders had no role in study design, data collection and analysis, decision to publish, or preparation of the manuscript.

### Grant Disclosures
The following grant information was disclosed by the authors:
Russian Science Foundation: 22-76-10051.

### Competing Interests
The authors declare that they have no competing interests.

### Author Contributions

- Olga V. Polenogova conceived and designed the experiments, performed the experiments, analyzed the data, prepared figures and/or tables, authored or reviewed drafts of the article, and approved the final draft.
- Natalia A. Kryukova conceived and designed the experiments, analyzed the data, prepared figures and/or tables, authored or reviewed drafts of the article, and approved the final draft.

- Tatyana Klementeva performed the experiments, prepared figures and/or tables, and approved the final draft.
- Anna S. Artemchenko performed the experiments, analyzed the data, prepared figures and/or tables, and approved the final draft.
- Alexander D. Lukin performed the experiments, analyzed the data, prepared figures and/or tables, and approved the final draft.
- Viktor P. Khodyrev performed the experiments, prepared figures and/or tables, and approved the final draft.
- Irina Slepneva performed the experiments, analyzed the data, authored or reviewed drafts of the article, and approved the final draft.
- Yana Vorontsova performed the experiments, analyzed the data, prepared figures and/or tables, authored or reviewed drafts of the article, and approved the final draft.
- Viktor V. Glupov conceived and designed the experiments, analyzed the data, authored or reviewed drafts of the article, and approved the final draft.

### Data Availability

The raw measurements are available in the Supplemental File.

### Supplemental Information

Supplemental information for this article can be found online at http://dx.doi.org/10.7717/peerj.18259#supplemental-information.

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
