# Peer review of "The influence of inactivated entomopathogenic bacterium Bacillus thuringiensis on the immune responses of the Colorado potato beetle"

_PeerJ, doi:10.7717/peerj.18259_

## Round 0.1 · original submission · Major Revisions

Dear Dr. Polenogova and colleagues:

Thanks for submitting your manuscript to PeerJ. I have now received three independent reviews of your work, and as you will see, the reviewers raised some concerns about the manuscript. Despite this, these reviewers are optimistic about your work and the potential impact it will have on research studying microbe-induced immune responses of the Colorado potato beetle. Thus, I encourage you to revise your manuscript, accordingly, considering all the concerns raised by the three reviewers.

While the concerns of the reviewers are relatively minor, this is a major revision to ensure that the original reviewers have a chance to evaluate your responses to their concerns. There are many suggestions, which I am sure will greatly improve your manuscript once addressed.

Good luck with your revision,

Best,

-joe

·

Basic reporting

The article structure conforms to PeerJ standards and disciplinary norms . The figures are relevant, well-labelled and described. Raw data have been supplied in accordance with PeerJ policy. This study addresses an important aspect of entomopathogenic bacteria's interaction with insect hosts. Minor revisions are required to make the manuscript to be more clear and accessible to worldwide readers. Consider revising the document based on the specific feedback to enhance clarity and readability. Please see following comments for revisions.
1. Line 26-28; Abstract background: “This can cause an imbalance in the gut microbial community, leading to increased mediator synthesis and changes in the host's physiological state. Additionally, enzyme synthesis in the gut may increase.” These sentences are not clear. What does "mediator synthesis" mean? Are you referring to detoxification enzymes or some other type of enzymes?
2. Line 37-38: 1.4–2.2-fold increase, please specify weather these values are based on comparison with control levels or with active Bt bacteria?
3. Line 35: Results: “12 hrs post-inoculation “ change it as “after 12 hrs post-inoculation”
4. Revise the sentence “This leads to the need to detoxify toxic products of the microbiota resulting from recognition of the 'putative' pathogen and an overabundance of secondary metabolites” for ease of understanding.
5. The expression of enzyme units used in the manuscript needs to be crosschecked throughout the manuscript. You may refer Toxicol Rep. 2020; 7: 1428–1437.
6. Line 428: Delete the colon after "DOI". Line 442: The journal name should be in italics. Lines 458-459: The journal name should be in the same format. All references need to be cross-checked for uniformity.
7. Line 391: change it to Stączek et al., 2023”
8. Line 305-206, It would be appropriate to include a possible explanation for this phenomenon.
9. Line 396 to 407, Conclusions: The conclusions seem overly ambitious, as the authors did not conduct experiments on gut microbial community dynamics or hot-pathogen receptor-related studies or what type of metabolites you are refereed to? I suggest the authors to be more precise in presenting their findings. Few of these points can be transferred to Discussion section.
10. 387-391: Simplify this paragraph.
11. When you inactivate the pathogen with autoclaving, it is anticipated that the pathogen-associated molecular patterns (PAMPs) will be disintegrated, degraded, or undergo conformational changes. Consequently, there will be no interaction between the host's pattern recognition receptors (PRRs) and the pathogen's PAMPs, which would result in no elicitation of immune mechanisms in the host insect. Therefore, this method may not work for immune priming against entomopathogens in host insects.

Experimental design

The experiments were properly conducted, with appropriate statistical tools employed to determine the significance of the data. The research question is well-defined, relevant, clearly stating how the research fills an identified knowledge gap. Relevant information is provided to support the study's context and objectives.

Validity of the findings

The results section is informative and highlights key differences and similarities between the effects of inactivated and active Bt bacteria on CPB larvae, offering valuable insights into the immune responses of these insects. Conclusions section needs to be improved. Please see "basic reporting" comments for revisions

Additional comments

The manuscript entitled “The Influence of Inactivated Entomopathogenic Bacteria Bacillus thuringiensis on the Immune Responses of Colorado Potato Beetle” is well-structured and provides a clear distinction between the effects of inactivated and active Bacillus thuringiensis (Bt) on the immune responses of Leptinotarsa decemlineata Say. The background section effectively sets the context for the research, explaining the relevance and importance of studying these effects. The experimental methodologies are detailed and encompass a comprehensive range of physiological and biochemical parameters, ensuring thorough examination. The results section is informative and highlights key differences and similarities between the effects of inactivated and active Bt bacteria on CPB larvae, offering valuable insights into the immune responses of these insects. However, When you inactivate the pathogen with autoclaving, it is anticipated that the pathogen-associated molecular patterns (PAMPs) will be disintegrated, degraded, or undergo conformational changes. Consequently, there will be no interaction between the host's pattern recognition receptors (PRRs) and the pathogen's PAMPs, which would result in no elicitation of immune mechanisms in the host insect. Therefore, this method may not work for immune priming against entomopathogens in host insects.

Reviewer 2 ·

Basic reporting

Authors reported on the effects of active and heat-killed Bt on several physiological and immunological parameters of the Colorado potato beetle. Overall, the manuscript is well written and organized, and it can represent an useful integration of the available data on the insects' immunity response to this bacterium.

Experimental design

Authors should add more details on the statistical analysis. Captions of Figure 1 should include information on how many samples were analyzed for each group (N=?). In the other figures it should be reported what kind of test was conducted: particularly, are all Kruskall-wallis as described in the Statistical Analysis section? Have time points been analyzed separately?. Moreover, in Figure 3 asterisks denote differences from the control group; however, if Kruskall-wallis test was used, followed by Dunn’s test, differences should be denoted by letters, because comparisons should be performed between all means/medians.

Validity of the findings

The manuscript could be improved by more clearly stating the scopes of the work in the introduction and by further discussing the putative roles of other microbial associates and their secondary metabolites, which have not been explored by the authors. Particularly, more relevant literature should be considered in introducing/discussing aspects related the role by midgut microbiota in immunity towards Bt (e.g. doi.org/10.1073/pnas.152174111).

Additional comments

The following minor corrections are required:
Line 59: the statement that 'CPB is resistant to .... genetically modified crops' is incorrect. Do you mean that CPB is able to attack transgenic plants?
Line 70: 'Cyt' has not been previously introduced.
Line 103: use the full name Galleria mellonella.
Line 141: full name and abbreviation CPB of Leptinotarsa decemlineata (Say) (CPB) have already been defined in the introduction; use just L. decemlineata here.
Lines 158, 161 and 163: correct to 'were'.
Line 171: correct to 'hemocytes'.
Line 308: use 'THC' only (definition has been previously provided).
Lines 340-341: correct to 'when infected with both active and inactivated bacteria'.
Line 351: correct 'moths' to 'larvae'.
Line 352: use abbreviated form 'B. thuringiensis'.
Line 388: correct to '...microbiota secondary...'.
Line 403: use the abbreviated form 'CPB'.
Caption of Figure 4 is truncated to “entomop”.

·

Basic reporting

The manuscript by Polenogova and colleagues is written in clear English, ensuring that the content is comprehensible to a broad audience. The background summarizes well the key message of the manuscript, i.e., the one-to-one comparison of Colorado potato beetle (CBP) treated with heat inactivated vs. vital B. thuringiensis using assays involving total hemocyte count, phenoloxidase and glutathione S-transferase, esterases, catalases, peroxidases, etc. activity and ROS formation.
I had difficulties to follow your introduction; it does not seem to follow a stringent line of argumentation to present the basic idea of this manuscript. I suggest that you include your hypothesis in the introduction, what you would expect in the comparison of inactivated B. thuringiensis vs. active B. thuringiensis cells. Comparing heat inactivated with alive bacteria fed CBP typically aims to discern the contributions of effects on the host to bacterial viability and toxin production, and host immune response of the host to the different treatments. For example, the inactivated bacteria might still trigger an immune response in the insect, but it would be different from the response to live bacteria that you want to discern. You are stressing the importance of ROS in the innate immune repertoire of insects, I would suggest, that you state that most of the publication refer to Drosophila melanogaster and that you want to study the role of ROS etc. in the CBP model.
The literature is well referenced and most of the times relevant. Maybe the damage inflicted by CPB can be cited with FAO statistics, see “Food and Agriculture Organization of the United Nations FAO STAT. Available online: http://www.fao.org/faostat/en/#data/QC/visualize (accessed on 31 March 2020).” General information and a good publication for citation about CPB can be found in “Radcliffe, E. B., and A. Lagnaoui. "Pests and diseases." Potato Biology and Biotechnology: Advances and Perspectives, 1st ed.; Vreugdenhil, D., Bradshaw, J., Gebhardt, C., Govers, F., Taylor, MA, MacKerron, DK, Ross, HA, Eds (2007): 545-554.”
Figure are relevant, of sufficient quality, well labelled and described. As recommendation, please consider to change the font of the figures to a sans serif type to enhance the uniformity of the manuscript.

Experimental design

To my opinion, the original primary research is within the scope of the PeerJ journal. The research question is well defined and relevant, since such experiments fills a gap in the understanding of the relevance of alive or dead B. thuringiensis to the host response of CBP. The experimental setup is described sufficiently in the methods section with adequate replications, the assays are robust and follow high technical standards. I would recommend to add in the Bacteria part (L136-following) that the negative control of the survival assay consists of saline solution and remove the information in the Fig. legend 1.

Validity of the findings

The impact of your manuscript is for me clear, it represents a unique approach to study the CBP host response to alive vs. dead B. thuringiensis with biological and biochemical means. To my best knowledge, this is the first time that this scientific question was raised. In the discussion, you could elaborate on the potential of B. thuringiensis as base for biological control; here, you could also discuss the finding, that the strain you are using, the Bacillus thuringiensis var. tenebrionis is only effective in controlling young stages of CBP, i.e., the first and second instars, according to Pucci, 1992 (Pucci, C. (1992), Biological control of Potato beetle, Leptinotarsa decemlineata Say (Col., Chrysomelidae) in Northern Latium, Central Italy†. Journal of Applied Entomology, 113: 194-201. https://doi.org/10.1111/j.1439-0418.1992.tb00653.x).
All underlying data have been provided; they are robust and statistically sound. However, the conclusions need to be reworked, they are not linked consistently to the original research question, see comments below.

Additional comments

General comments:
For the discussion part, I like to see a more detailed view on the effects of living bacteria. The live bacteria might trigger a more complex immune response, potentially involving both the toxins and bacterial metabolites, and also be responsible for up- or downregulating immune related gene expression compared to the heat inactivated treatment. A discussion, how inactivated bacteria could still trigger a response, should be discussed, for example, can cry toxin be heat-stable and induce an effect? In future projects, would it be useful to use a transcriptomic approach on the base of this manuscript? You might discuss this in the light of a paper dealing with CBP transcriptomic response to the insecticide Spinosad, see Bastarache et al., 2020 (Bastarache P, Wajnberg G, Dumas P, Chacko S, Lacroix J, Crapoulet N, Moffat CE, Morin P Jr. Transcriptomics-Based Approach Identifies Spinosad-Associated Targets in the Colorado Potato Beetle, Leptinotarsa decemlineata. Insects. 2020 Nov 21;11(11):820. doi: 10.3390/insects11110820. PMID: 33233355; PMCID: PMC7700309).
For the THC part, it would be nice to discuss this in light of a recent manuscript by Prabu and colleagues (Prabu S, Jing D, Jurat-Fuentes JL, Wang Z, He K. Hemocyte response to treatment of susceptible and resistant Asian corn borer (Ostrinia furnacalis) larvae with Cry1F toxin from Bacillus thuringiensis. Front Immunol. 2022 Nov 17;13:1022445. doi: 10.3389/fimmu.2022.1022445. PMID: 36466886; PMCID: PMC9714555.), dealing with circulating hemocytes and Cry toxin treatment in the Asian corn borer Ostrinia furnacalis.
Minor comments:
L75-77: Ref. missing.
L110: Buchon, Broderick & Lemaitre, 2013 refer to results from Drosophila melanogaster, second reference refers more to AMPs, I am not sure, what should be cited here in this context.
L340: hemolymph instead of lymph.
L347: hemolymph instead of lymph.
L380-386: seems to be a repetition of results without discussion, please rephrase this passage.
L392-394: Last sentence a bit too speculative, this has not been demonstrated with the presented data.
L397: gut microbial community has not been studied in this manuscript, please rephrase.

---

## Round 0.2 · accepted · Accept

Dear Dr. Polenogova and colleagues:

Thanks for revising your manuscript based on the concerns raised by the reviewers. I now believe that your manuscript is suitable for publication. Congratulations! I look forward to seeing this work in print, and I anticipate it being an important resource for groups studying microbe-induced immune responses of the Colorado potato beetle. Thanks again for choosing PeerJ to publish such important work.

Best,

-joe

·

Basic reporting

The manuscript has been revised according to the suggestions and is now accepted in its current form

Experimental design

Appropriate

Validity of the findings

Appropriate

Additional comments

The manuscript has been revised according to the suggestions and is now accepted in its current form